

# A practical guide to build *de-novo* assemblies for single tissues of non-model organisms: the example of a Neotropical frog

Santiago Montero-Mendieta[1], Manfred Grabherr[2], Henrik Lantz[2], Ignacio De la Riva[3], Jennifer A. Leonard[1], Matthew T. Webster[4] and Carles Vilà[1]

[1] Conservation and Evolutionary Genetics Group, Department of Integrative Ecology, Doñana Biological Station (EBD-CSIC), Consejo Superior de Investigaciones Científicas, Seville, Spain

[2] Department of Medical Biochemistry and Microbiology, National Bioinformatics Infrastructure Sweden (BILS), Uppsala Universitet, Uppsala, Sweden

[3] Department of Biodiversity and Evolutionary Biology, Museo Nacional de Ciencias Naturales, Consejo Superior de Investigaciones Científicas, Madrid, Spain

[4] Department of Medical Biochemistry and Microbiology, Science for Life Laboratory, Uppsala Universitet, Uppsala, Sweden

Corresponding authors
Santiago Montero-Mendieta, santiago.montero@ebd.csic.es
Carles Vilà, carles.vila@ebd.csic.es

## ABSTRACT

Whole genome sequencing (WGS) is a very valuable resource to understand the evolutionary history of poorly known species. However, in organisms with large genomes, as most amphibians, WGS is still excessively challenging and transcriptome sequencing (RNA-seq) represents a cost-effective tool to explore genome-wide variability. Non-model organisms do not usually have a reference genome and the transcriptome must be assembled *de-novo*. We used RNA-seq to obtain the transcriptomic profile for *Oreobates cruralis*, a poorly known South American direct-developing frog. In total, 550,871 transcripts were assembled, corresponding to 422,999 putative genes. Of those, we identified 23,500, 37,349, 38,120 and 45,885 genes present in the Pfam, EggNOG, KEGG and GO databases, respectively. Interestingly, our results suggested that genes related to immune system and defense mechanisms are abundant in the transcriptome of *O. cruralis*. We also present a pipeline to assist with pre-processing, assembling, evaluating and functionally annotating a *de-novo* transcriptome from RNA-seq data of non-model organisms. Our pipeline guides the inexperienced user in an intuitive way through all the necessary steps to build *de-novo* transcriptome assemblies using readily available software and is freely available at: https://github.com/biomendi/TRANSCRIPTOME-ASSEMBLY-PIPELINE/wiki.

## INTRODUCTION

The word "genomics" refers to the study of the complete set of genes and gene products in an individual. With the ongoing reduction of costs, this is frequently achieved through the use of high-throughput sequencing technologies (*Reuter, Spacek & Snyder, 2015*).

The "genomics era" formally started after the Human Genome Project (HGP) was first published in 2001 (*Lander et al., 2001*). Since then, genomics has drastically changed the way that we understand and study the genetic features of living organisms. Mainly due to novel gene discovery, genomics has proved useful in many fields, such as molecular medicine (*Giallourakis et al., 2005*), molecular anthropology (*Destro-Bisol et al., 2010*), social sciences (*McBride et al., 2010*), evolutionary biology (*Wolfe, 2006*) and biological conservation (*McMahon, Teeling & Höglund, 2014*), among others. Nowadays, a main use of genomics is to profile genomes, transcriptomes, proteomes, and metabolomes (*Schuster, 2008*). Genomics has also proved highly informative in elucidating evolutionary history of species and, for example, has enabled finding genes that could explain the variation in beak size within and among species of Darwin's finches, in addition to providing new insights into the evolutionary history of these birds (*Lamichhaney et al., 2015*; *Lamichhaney et al., 2016*).

At the time of writing this article (January 2017), 8,951 genomes had been completely sequenced according to the Genomes OnLine Database (GOLD) (https://gold.jgi.doe.gov) (*Mukherjee et al., 2017*). These genomes include mainly unicellular organisms (4,958 bacteria; 240 archaea) and viruses (3,473) due to their small genome size. Eukaryote organisms usually have larger genomes and the sequencing effort to fully sequence them is much larger. Only 280 eukaryote genomes have been completed, most of them belonging to model organisms (i.e., species that have been widely studied because of particular experimental advantages or biomedical interest). However, the difficulties associated with the assembly of large genomes have resulted in very few of these being fully sequenced. Among terrestrial vertebrates, amphibians have the largest genome sizes. The average genome size of frogs is 5.0 gigabases (Gb), while the fire salamander (*Salamandra salamandra*) genome averages 34.5 Gb (*Gregory et al., 2007*). For this reason, few genomics studies on amphibians have been carried out so far. To date, only the genomes of three frogs of reduced genome size, *Xenopus tropicalis* (1.5 Gb; *Hellsten et al., 2010*), *Xenopus laevis* (2.7 Gb; *Session et al., 2016*) and *Nanorana parkeri* (2.3 Gb; *Sun et al., 2015*), have been sequenced and published, in contrast to the larger number of genomes of reptiles (10), birds (53) and mammals (43). Due to the difficulties to obtain reference genome sequences for species with large genome sizes, reduced representation approaches are a cost-effective way to obtain information on genome-wide variability. For non-model organisms in which whole genome sequencing (WGS) is not feasible, transcriptome (e.g., *Geraldes et al., 2011*; *De Wit, Pespeni & Palumbi, 2015*) or exome (*Lamichhaney et al., 2012*) sequencing are commonly used as a reduced representation of the genome.

In amphibians, 24 transcriptomes from 19 species are currently available in the Transcriptome Shotgun Assemblies (TSA) database (https://www.ncbi.nlm.nih.gov/genbank/tsa/, January 2017), highlighting the importance of RNA sequencing (RNA-seq) for genomic studies in this group. RNA-seq is more affordable than whole genome sequencing and has rapidly become the preferred method for cataloguing and quantifying the complete set of transcripts or messenger RNA for a specific tissue, developmental stage or physiological condition (*Wang, Gerstein & Snyder, 2009*). Nowadays, RNA-seq has a wide variety of uses but the core analyses include transcriptome profiling, differential

gene expression and functional profiling (*Conesa et al., 2016*). As transcriptome assembly becomes more common for non-model and poorly known organisms, we expect it will become a more popular tool also in phylogenomics as well as in demographic and population structure inference. However, what kind of RNA-seq data analysis to be performed depends on the species of interest and the research goals. For model organisms and their close relatives, RNA-seq data is analyzed by mapping reads to a reference genome. By contrast, most non-model organisms do not have a reference genome from a sufficiently closely related species, and the transcriptome must be assembled *de-novo* (*Martin & Wang, 2011*). Many bioinformatics tools to build a *de-novo* transcriptome are now available, yet contrasting opinions about the steps to follow may be disorienting. Some extremely simple pipelines have been developed to automatize the process (e.g., TRUFA; *Kornobis et al., 2015*), but this may limit the flexibility of the different pieces of software that have been integrated.

Here, we present the transcriptome profile for *Oreobates cruralis*, a direct-developing frog species from the Amazonian regions of Bolivia and Peru. To date, this is the first transcriptome available for a South American amphibian. We also present a simple pipeline for pre-processing, building and functionally annotating a *de-novo* transcriptome from RNA-seq data of non-model organisms using available software.

## METHODS

### Study model and sample collection

The genus *Oreobates* Jiménez de la Espada, 1872 (Anura: Craugastoridae) is a poorly studied clade of New World direct-developing frogs (Terrarana) distributed from the lower slopes of the eastern Andes into the upper Amazon basin, encompassing from southern Colombia and western and central Brazil up to northern Argentina (*Jiménez de la Espada, 1872*). More than half of the 24 identified species have been described in the last decade and the species diversity in this genus is likely to be underestimated (*Köhler & Padial, 2016*). One of these species, *O. cruralis* (*Boulenger, 1902*) occurs in a wide range of elevations and habitats across Bolivia and Peru. Its distribution includes lowland Amazonian rainforests (approximate altitudinal range, from 100 to 600 m above sea level, m.a.s.l.), Yungas-montane Amazonian rainforests (600–2,500 m.a.s.l.), and inter-Andean dry valleys (1,300–3,000 m.a.s.l.) (*De la Riva et al., 2000*). However, little is known about its ecology and evolutionary history.

For this study we used tissue samples from a single individual of *O. cruralis,* sampled in Bolivia (Villa Tunari, Cochabamba, Bolivia; 345 m.a.s.l.; 16°59′01.4″S 65°24′30.16″W) on November 28th, 2013 as part of an unrelated project and deposited at the tissue collection of the Museo Nacional de Ciencias Naturales (MNCN-CSIC) in Madrid, Spain (MNCN/ADN:65263; Colección Boliviana de Fauna, CBF 7268). Unfortunately, the specimen was unsexed. Samples of five tissues (intestine, liver, spleen, heart and skin) were isolated and preserved in Nucleic Acid Preservation (NAP) buffer (*Camacho-Sanchez et al., 2013*) at the time of sampling and were later kept at −80 °C. No other tissues were preserved in buffers suitable for RNA preservation.

## Ethics statement

The original field surveys that led to sampling the studied specimen were approved by the Dirección General de la Biodiversidad, Ministerio de Medio Ambiente y Agua, La Paz, Bolivia (Approval number: MMAyA-VMA-DGBAP No 1592/12), and were supported and approved by the Spanish Government (Ministerio de Economía y Competitividad, Project numbers: CGL2011-30393, awarded to IDlR).

## Transcriptome sequencing

We extracted whole RNA for each tissue using the RNeasy Protect Mini Kit (Qiagen). RNA quality was evaluated with RNA ScreenTape on TapeStation by Agilent. Due to poor RNA quality, two tissues were discarded (skin and heart), thus only RNA extracts from intestine, liver and spleen were used (RIN, RNA integrity number, scores of 6.2, 7.3 and 7.1, respectively). Sequencing libraries were prepared and sequenced by the SNP&SEQ Technology Platform (Uppsala University) from 1 µg total RNA using the TruSeq stranded mRNA library preparation kit (Illumina Inc.) and including poly-A selection. The library preparation was performed according to the manufacturers' protocol. The quality of the libraries was evaluated using the Agilent Technologies TapeStation and a DNA 1000-kit Screen Tape. The adapter-ligated fragments were quantified by qPCR using the Library quantification kit for Illumina (KAPA Biosystems) on a StepOnePlus instrument (Applied Biosystems/Life technologies) prior to cluster generation and sequencing. A 14 pM solution of RNA was subjected to cluster generation and paired-end sequencing with 125 bp (base pair) read length on a HiSeq2500 instrument (Illumina Inc.) using the v4 chemistry according to the manufacturer's protocols.

## RNA-seq data analysis

The overall pipeline is summarized in Fig. 1 and our practical guide is available at: https://github.com/biomendi/TRANSCRIPTOME-ASSEMBLY-PIPELINE/wiki. Briefly, the first step after obtaining the raw sequence data in FASTQ format was to perform a preliminary quality control analysis with FastQC (http://www.bioinformatics.babraham.ac.uk/projects/fastqc/). FastQC delivers quality metrics that are useful to identify if the data requires initial pre-processing before the transcriptome assembly. The pre-processing stage included three steps: first, removal of possible ribosomal RNA (rRNA) contamination; second, trimming low quality bases and PCR adapters; third, normalization to remove large excess of reads corresponding to moderately and highly expressed transcripts. Pre-processing is not always needed but it is highly recommendable to improve assembly quality. Once the data was pre-processed, a quality control was performed again and then, clean normalized reads were *de-novo* assembled in absence of a reference genome. Subsequent analyses depend on the study goals. In our case, transcripts were functionally annotated using various databases to obtain a transcriptome profile. All steps are described in further detail in the following paragraphs.

We filtered raw FASTQ reads using SORTMERNA-v2.1 (*Kopylova, Noé & Touzet, 2012*) against 8 default rRNA databases (SILVA 16S bacteria, SILVA 16S archaea, SILVA 18S eukarya, SILVA 23S bacteria, SILVA 23s archaea, SILVA 28S eukarya, Rfam 5S

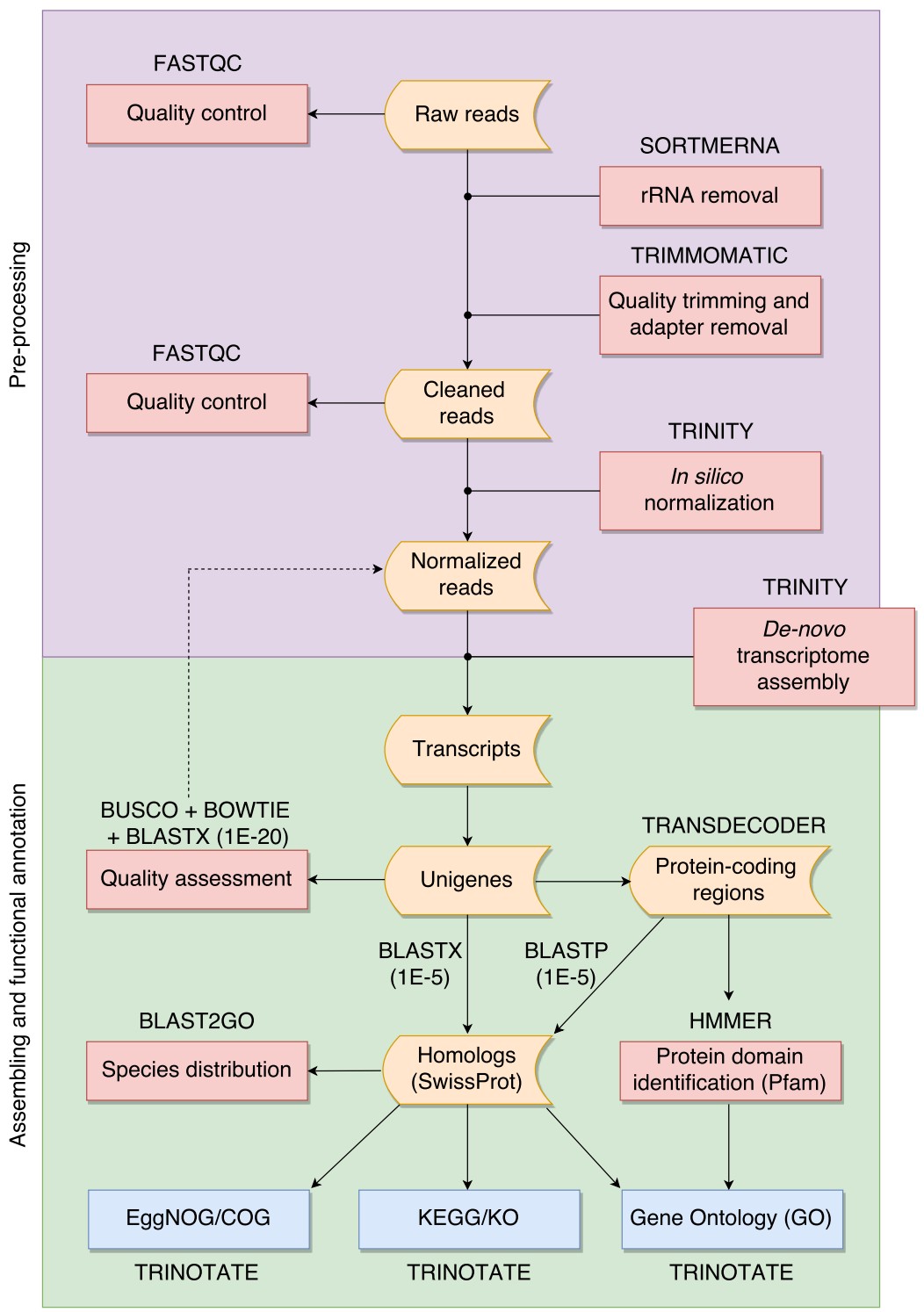

**Figure 1** **Overall pipeline for the annotation of RNA-seq data.** Boxes with curved sides represent sequence datasets. Red boxes represent analyses, and the software used for those analyses is indicated outside the box. Reference databases are indicated as blue boxes.

archaea/bacteria, Rfam 5.8S eukarya) to remove rRNA. Then, we used TRIMMOMATIC-v0.32 (*Bolger, Lohse & Usadel, 2014*) to trim adaptors and sequences with Phred quality score <20. We normalized cleaned data of each tissue using the *in-silico* normalization utility included in the TRINITY-2.2.0 package (*Grabherr et al., 2011*). Normalization is useful for large RNA-seq data sets (>300 million paired-end reads) because it will remove over-expressed transcripts, thus lowering computing memory consumption and speeding up the assembly process (*Haas et al., 2013*). We merged the resulting data for the three tissues into a single dataset and normalized again to remove redundant sequences that could have been obtained from several tissues. We used TRINITY (*Grabherr et al., 2011*) to *de-novo* assemble normalized reads into contigs. This resulted in a large number of transcripts, much higher than the expected number of genes, likely because of alternative splicing. To avoid redundant transcripts, we kept the longest isoform for each "gene" identified by TRINITY (unigene) using the "get_longest_isoform_seq_per_trinity_gene.pl" utility in TRINITY. Thus, each unigene represented a collection of expressed sequences (i.e., transcripts) that apparently came from the same transcription locus, representing a putative gene. This set of unigenes was kept for downstream analyses.

We evaluated the quality of the assembly and the transcript contiguity in terms of read representation by mapping normalized reads back to the set of unigenes using BOWTIE-1.1.2 (*Langmead et al., 2009*). We assessed the assembly completeness in terms of gene content using BUSCO-v1 (*Simao et al., 2015*) by searching the unigenes for the presence or absence of conserved orthologs in the tetrapoda-odb9 database (http://busco.ezlab.org/datasets/tetrapoda_odb9.tar.gz) that represents a collection of 3,950 single-copy tetrapoda orthologs. We also mapped with $E$-value $\leq$ 1E−20 the unigenes to the SwissProt database (ftp://ftp.ebi.ac.uk/pub/databases/uniprot/) and to the Western clawed frog (*Xenopus tropicalis*) proteome (http://ftp.ensembl.org/pub/release81/fasta/xenopus_tropicalis/pep/Xenopus_tropicalis.JGI_4.2.pep.all.fa.gz) using BLASTX (searches within a protein database using a translated nucleotide query) included in the NCBI-BLAST-2.4.0+ package (*Altschul et al., 1990*). The SwissProt is a curated protein sequence database aimed to provide a high level of annotation (e.g., the description of the function of a protein), a minimal level of redundancy and high level of integration with other databases (*Bairoch & Apweiler, 2000*). There is no perfect $E$-value cut-off in BLAST, but the smaller the most reliable the match. We used orthologous proteins found in SwissProt and *X. tropicalis* to assess completeness as described by *Haas et al. (2013)*.

We predicted protein-coding regions in the unigenes based on the most likely longest-ORF using TransDecoder-v3 (*Haas et al., 2013*). In order to annotate the sequences we compared them to public databases compiled for different purposes. We searched homolog sequences for the predicted proteins using BLASTP (search protein database using a protein query) with $E$-value $\leq$ 1E−5 to the SwissProt database. We also used BLASTX with $E$-value $\leq$ 1E−5 to search homolog sequences for the unigenes compared to the SwissProt database. In both cases, BLASTP and BLASTX, we only kept top-hit matches. We used BLAST2GO (*Conesa et al., 2005*) to detect the species distribution of the top BLASTX results within the SwissProt database. We identified

protein domains using HMMER-3.1b2 (*Finn, Clements & Eddy, 2011*) to the Pfam-A database (ftp://ftp.ebi.ac.uk/pub/databases/Pfam/). Homologous proteins found in the SwissProt database were used to retrieve functional annotation comments from the GO (*Gene Ontology; Ashburner et al., 2000*), EggNOG (*Evolutionary Genealogy of Genes: Non-supervised Orthologous Groups; Powell et al., 2012*) and KEGG (*Kyoto Encyclopedia of Genes and Genomes; Kanehisa et al., 2012*) databases using TRINOTATE-v.3 (https://trinotate.github.io). The software also searched GO terms in Pfam results and in the combined results of homology search via SwissProt and Pfam. At the time of conducting this study, TRINOTATE was built around specific releases of SwissProt and Pfam databases (available at https://data.broadinstitute.org/Trinity/Trinotate_v3_RESOURCES/). We used BLAST2GO to categorize the annotated GO terms in the combined results of SwissProt and Pfam searches. EggNOG annotations were filtered to keep COGs (Clusters of Orthologous Groups) and those were categorized using the current version of the COG database (ftp://ftp.ncbi.nih.gov/pub/COG/COG2014/data). KEGG annotations were filtered to keep KOs (KEGG orthology) and those were categorized using the tool "Reconstruct Pathway" (http://www.kegg.jp/kegg/tool/map_pathway.html).

## Data availability

Raw RNA-seq data in FASTQ format has been deposited at the NCBI Sequence Read Archive database (SRA) under the accession SRP106442. The transcriptome assembly in FASTA format has been deposited at DDBJ/EMBL/GenBank under the accession GFNJ00000000. The quality of the assembly was examined through the NCBI contamination screen. The screen found five sequences to exclude, 105 sequences with locations to mask/trim and six potentially duplicated sequences (with three distinct checksums). As a result, the uploaded information contained 422,970 sequences (188,369,677 bp) rather than the initial 422,999 sequences (188,399,293 bp). All the data is available at NCBI BioProject under the accession PRJNA384528.

# RESULTS AND DISCUSSION

## RNA sequencing and transcriptome assembly

A summary of the RNA-seq data and transcriptome assembly is presented in Table 1. Illumina RNA sequencing for three tissues of *O. cruralis* in an Illumina HiSeq2500 instrument produced a total of almost $523 \times 10^6$ raw reads (intestine: $194 \times 10^6$; liver: $189 \times 10^6$; spleen: $140 \times 10^6$). Of those, 81.47% were kept after the pre-processing stage ($426 \times 10^6$). The number of reads was further reduced to 6.97% after *in silico* normalization prior to assembly ($36 \times 10^6$). This highlights the importance of normalization to remove over-expressed transcripts in RNA-seq data. A total of 550,871 transcripts were obtained after *de-novo* transcriptome assembly. This large number of transcripts is not too surprising, both in terms of RNA-seq assembly as well as given the species and its likely large genome (see genome size for closely related genera at: http://www.genomesize.com/). First, transcriptome assemblies often include incompletely spliced introns, orphaned UTRs, read through off of the 3′ ends, spuriously transcribed regions, active transposable elements, etc., so the number of assembled transcripts typically exceeds the expected number of

**Table 1** Summary of the transcriptome data assembly for *Oreobates cruralis*.

| Prior to *de-novo* transcriptome assembly | |
|---|---|
| Length of raw reads (bp) | 125 |
| Total number of raw reads | 522,877,358 |
| Total number of clean reads | 426,003,462 |
| Total number of normalized reads | 36,428,858 |
| After *de-novo* transcriptome assembly | |
| Total number of all transcripts/unigenes | 550,871/422,999 |
| GC-content of all transcripts/unigenes (%) | 45.88/45.39 |
| Total length of all transcripts/unigenes (bp) | 299,133,111/188,399,293 |
| N50 length of all transcripts/unigenes (bp) | 731/467 |
| Mean length of all transcripts/unigenes (bp) | 543/445 |
| Median length of all transcripts/unigenes (bp) | 309/290 |

protein coding genes by an order of magnitude. Second, large genomes tend to have large transcriptomes. In the axolotl (*Ambystoma mexicanum*) the transcriptome assembly had $-1.5 \times 10^6$ transcripts that clustered into $-1.3 \times 10^6$ putative genes (unigenes), and of those, 110,000 mapped to 30,000 SwissProt genes (*Bryant et al., 2017*). It is possible that these large genomes include a large number of repetitive sequences transcribed, which makes assembly more difficult and results in more fragmentation, especially when using diginorm (as in TRINITY) or any other *in silico* normalization. In *O. cruralis* the 550,871 transcripts clustered into 422,999 unigenes. This difference in number is likely because of alternatively spliced isoforms derived from paralogous genes (*Wang et al., 2014*). However, this will need to be confirmed with new amphibian genomes as they become available. Unigenes in the transcriptome of *O. cruralis* had an average GC content of 45.39%, which is very similar to other amphibians, such as the axolotl (*A. mexicanum* 45.56%; *Hall, Eisthen & Williams, 2016*), the green toad (*Bufotes viridis* 46.83%; *Gerchen et al., 2016*) and the common frog (*Rana temporaria* 44%; *Price et al., 2015*). The size of the unigenes in *O. cruralis* ranged from 201 to 16,804 bp with a mean length of 445 bp and a N50 length of 467 bp (Table 1; Fig. 2). The N50 value indicates that half of the transcriptome unigenes were at least 467 bp in length. The N50 length has been proposed as an estimator of genome assembly contiguity, since better assemblies will result in longer contigs (*Li et al., 2014*; *Simpson, 2014*). However, in transcriptome data this measure can be highly misleading because it does not assess assembly completeness in terms of read representation or gene content (*Simao et al., 2015*).

## Transcriptome quality assessment

The set of assembled unigenes might not always perfectly correspond to all properly paired reads, as some unigenes might be built from just a portion of reads coming from the same transcription locus. When we evaluated assembly quality in terms of read representation, we found a high rate of reads that mapped back to unigenes (75.40%), thus confirming the presence of most of the initial reads in our final set of unigenes. When we evaluated the assembly completeness in terms of gene content, we found 2,830 complete orthologous

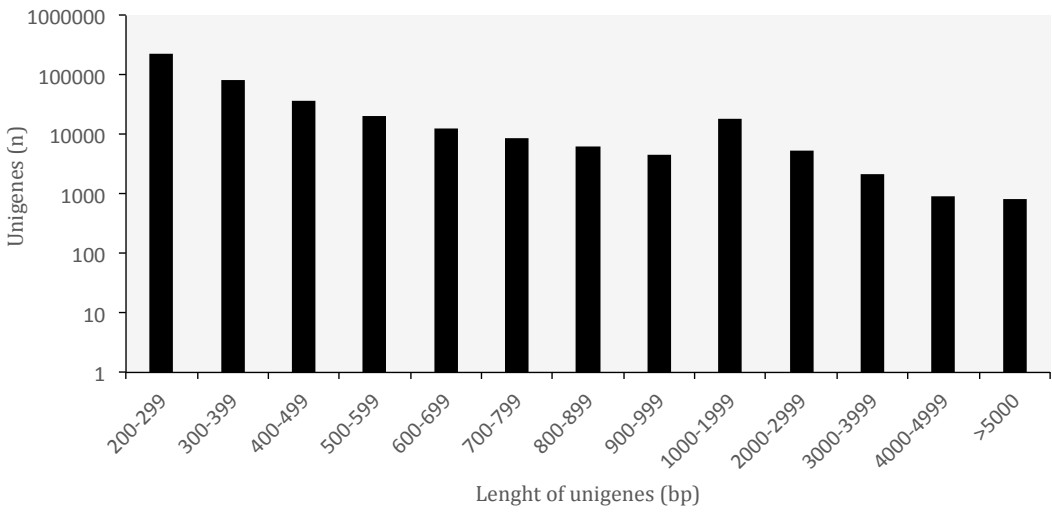

**Figure 2  Length distribution of unigenes from *Oreobates cruralis*.**

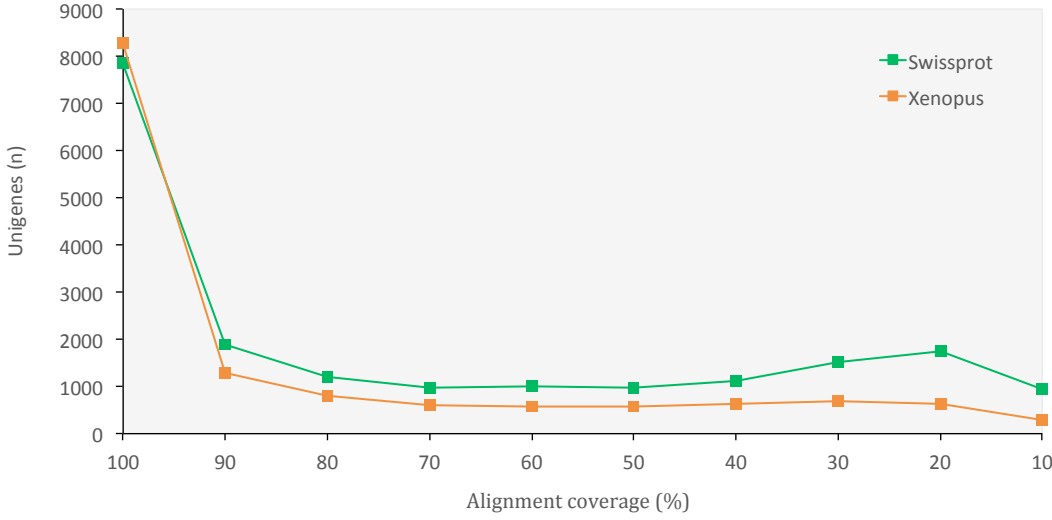

**Figure 3  Distribution of BLASTX alignment coverage for *O. cruralis* unigenes against SwissProt and *Xenopus* databases.** A high number of orthologous proteins in the databases fully or nearly fully corresponded (>80% coverage) to unigenes in *O. cruralis*.

genes (71.65%) out of the 3,950 genes available in the tetrapoda database (complete BUSCO hits). Of those, 2,501 were single-copy genes and 329 were duplicated genes. Only 462 (11.70%) of the genes in the database appeared fragmented and 658 (16.65%) were missing. We also obtained a high number of orthologous proteins in both the SwissProt and the *X. tropicalis* databases that fully matched (100% alignment coverage) or nearly fully corresponded (>80% alignment coverage) to unigenes in *O. cruralis* (Fig. 3). Altogether, the high number of complete (or nearly complete) orthologous matches across the different databases provides a valuable validation of the depth and completeness of the assembly process.

## Functional annotation of unigenes

Gene annotation consists of adding relevant biological information to coding regions of the genome and it was arguably the most relevant section of our pipeline, since it allowed us to describe and classify the content of the *O. cruralis* transcriptome. Functional annotation was based on BLAST searches to find homologous proteins within a reference database (e.g., SwissProt) and the collection of biological information from various sources (e.g., GO, KEGG, EggNOG or Pfam). We predicted a total of 45,466 protein-coding genes within the 422,999 unigenes using TansDecoder. After homology search using BLASTP, we found that 26,418 protein-coding genes in *O. cruralis* mapped to proteins in the SwissProt database. Search using BLASTX revealed a total of 54,425 unigenes that mapped to proteins in the *X. tropicalis* proteome and 47,349 unigenes that mapped to proteins in the SwissProt database. The relative low number of homologous proteins shared between *O. cruralis* and *X. tropicalis,* just 12.8% of all unigenes identified in *O. cruralis,* is a likely consequence of the very ancient divergence time between both species (estimated to be around 204 million years ago; http://www.timetree.org/). This ancient divergence implies a long time for the accumulation of mutations. However, the observation of a number of matches (54,425) that is larger than the total number of proteins in *X. tropicalis* (22,718) may suggest that many of them might be duplicates or unresolved splice variants among the unigenes of *O. cruralis*. The version of the SwissProt database used included a selection of 553,231 protein sequences from 13,379 species, and the top-hit species distribution showed that 32% (13,099) of the *O. cruralis* unigenes were homologs to human (*Homo sapiens*) proteins and 19% (7,661) to house mouse (*Mus musculus*) proteins (Fig. 4). The larger number of hits to mammals than to other amphibians is likely due to the uneven distribution of species in the SwissProt database, in which the top twenty species accumulate 21.5% of the entries. Still, amphibian species were highly represented in the assembly with 10% (3,909) of the *O. cruralis* unigenes having a highest match to *X. laevis* and 5% (1,893) to *X. tropicalis* proteins. When we retrieved the functional comments for the homologous proteins found in the SwissProt database, the number of annotated unigenes varied depending on the source that was used: a total of 45,885, 38,120, 37,349 and 23,500 unigenes were annotated for GO, KEGG, EggNOG and Pfam databases, respectively.

## Protein domain identification

Protein domains are preserved portions of proteins with tertiary structure that can act, evolve and exist independently of the rest of the protein chain (*Jacob, 1977*). Prediction of protein domains is an important step of transcriptome annotation since they provide insights in specific cellular functions that assist comparative genomics of domain families across species (*Ochoa, Llinás & Singh, 2011*). The Pfam database is a large collection of protein families that currently contains 16,303 families (Pfam v30.0). From the predicted 45,466 protein-coding genes in the transcriptome of *O. cruralis*, we identified 23,500 that are present in the Pfam-A database, consisting of 5,686 protein domain families. We found that the most common Pfam domain in the transcriptome of *O. cruralis* is the 'Zinc finger, C2H2 type' (961 hits; 4.09%). The C2H2 zinc finger proteins are very frequent in eukaryotic genomes (e.g., the human genome has 564 C2H2 zinc fingers; *Tadepally, Burger & Aubry,*

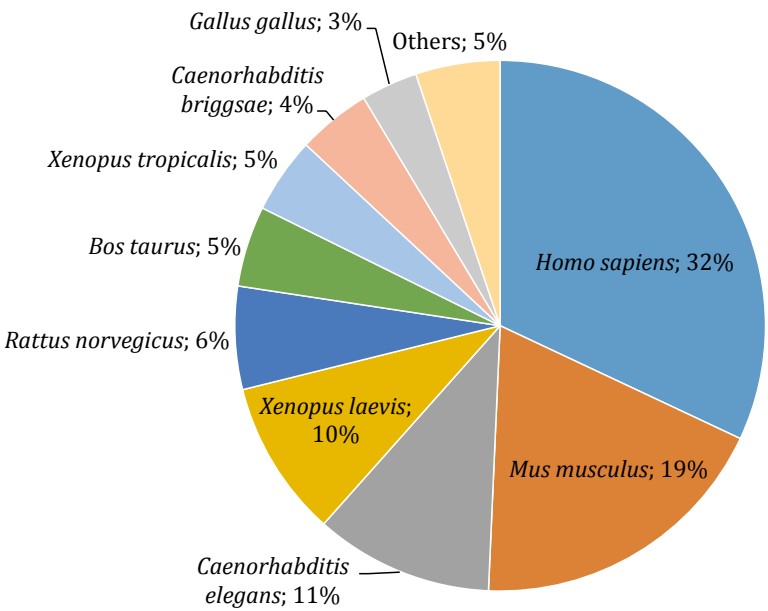

**Figure 4** Top-hit species distribution for unigenes from the transcriptome of *O. cruralis* in the Swiss-Prot database.

*2008*), and their functions are extraordinarily diverse, including DNA recognition, RNA packaging, transcriptional activation, regulation of apoptosis, protein folding and assembly, and lipid binding (*Laity, Lee & Wright, 2001*). Interestingly, this protein family was also reported as the most common for other amphibians, such as the green frog (*Lithobates clamitans*) and the Pacific tree frog (*Pseudacris regilla*) (*Robertson & Cornman, 2014*).

The 'WD domain, G-beta repeat' was the second most common Pfam domain in *O. cruralis* transcriptome (840 hits; 3.57%). The G protein family is involved in signal transduction from outside a cell to its interior (*Umbarger et al., 1992*), and in frog oocytes they are important in regulating the maturation process (*Kalinowski et al., 2003*). Another essential domain for frogs is the 'Protein kinase domain' that we found as the third more abundant (643 hits; 2.74%). This domain is supposed to play an important role in frogs in freezing tolerance during cold winters, likely inducing the transcription of antioxidant response genes (*Dieni & Storey, 2014*). Although freezing winters are not common within the current range of *O. cruralis*, the relative abundance of protein kinase domains could have been important in the evolutionary history of *Oreobates*, a genus that may have originated at high altitude in the Andes (*Padial, Chaparro & De la Riva, 2008*). It is also remarkably high the number of immunoglobulin-related domains found within the top 10 Pfam domains in the transcriptome of *O. cruralis* (1,066 hits; 4.54%) (Table 2). Immunoglobulin domains are involved in a wide range of functions, including cell–cell recognition, cell–surface receptors, muscle structure and immune system function (*Isenman, Painter & Dorrington, 1975*). In frogs, as in the Yunnan firebelly toad (*Bombina maxima*) (*Zhao et al., 2014*), these domains are essential for the regulation of immune responses, allowing them to survive in harsh environmental conditions. It is possible that tropical rainforests could host a large

**Table 2  Top 10 Pfam domains identified in the transcriptome of *O. cruralis*.**

| No | Pfam domain | Pfam ID | *N*-hits |
|---|---|---|---|
| 1 | Zinc finger, C2H2 type | PF00096.23 | 961 |
| 2 | WD domain, G-beta repeat | PF00400.29 | 840 |
| 3 | Protein kinase domain | PF00069.22 | 643 |
| 4 | Protein tyrosine kinase | PF07714.14 | 608 |
| 5 | C2H2-type zinc finger | PF13912.3 | 593 |
| 6 | C2H2-type zinc finger | PF13894.3 | 570 |
| 7 | Ankyrin repeat | PF00023.27 | 553 |
| 8 | Immunoglobulin I-set domain | PF07679.13 | 549 |
| 9 | Immunoglobulin domain | PF00047.22 | 517 |
| 10 | Leucine rich repeat | PF13855.3 | 482 |

diversity of potential pathogens imposing a positive selection on immunoglobulin-related domains in *Oreobates* frogs, but this hypothesis remains to be tested.

## Gene ontology

The Gene Ontology (GO) (http://geneontology.org/) is a standardized functional classification system aimed to describe gene and gene product attributes across species, using a controlled vocabulary (i.e., ontology terms). The GO classification comprises three domains: cellular component, molecular function and biological process. These domains have a hierarchical structure and a GO term can belong to different levels depending on the path followed and the number of steps between the term and the root (*Ashburner et al., 2000*). Using the combined results of a homology search via SwissProt and Pfam, we detected a total of 3,094,863 GO terms (19,407 unique) corresponding to 45,885 (10.85%) unigenes in the transcriptome of *O. cruralis*. This contrasts with previous studies that have reported that between 50 and 80% of the transcripts reconstructed from RNA-seq data can be annotated with GO terms (*Conesa et al., 2016*). However, the relatively low percentage of annotation may reflect the scarcity of amphibian sequences in the GO database, and therefore the presence of undetected novel transcripts. Still, the GO database produced the highest number of annotated unigenes compared to other sources, such as Pfam, KEGG or EggNOG. The largest number of GO terms corresponded to the category of "Biological Process" (49%) followed by "Cellular Component" (38%) and "Molecular Function" (13%). At ontology level-2, which represents the second most general category in the GO database, there were 65 different GO terms (Fig. 5). Within the "Biological Process" category, the most frequent GO terms were "cellular process" (35,730) and "single-organism process" (29,237). Within the "Molecular Function" category, unigenes were mainly associated to "binding" (32,275) and "catalytic activity" (18,023). Within the "Cellular Component" category, unigenes were mostly associated with "cell" (37,424) and "cell part" (37,293). These highly abundant GO terms are likely associated to genes involved in essential cell functions and metabolism regulation, since they describe very general terms. A similar distribution of GO terms was found in a comparative transcriptome study of seven anuran species (*Huang et al., 2016*). We found 185 unigenes with antioxidant activity, most of them with peroxidase activity (128). This number is relatively high compared to

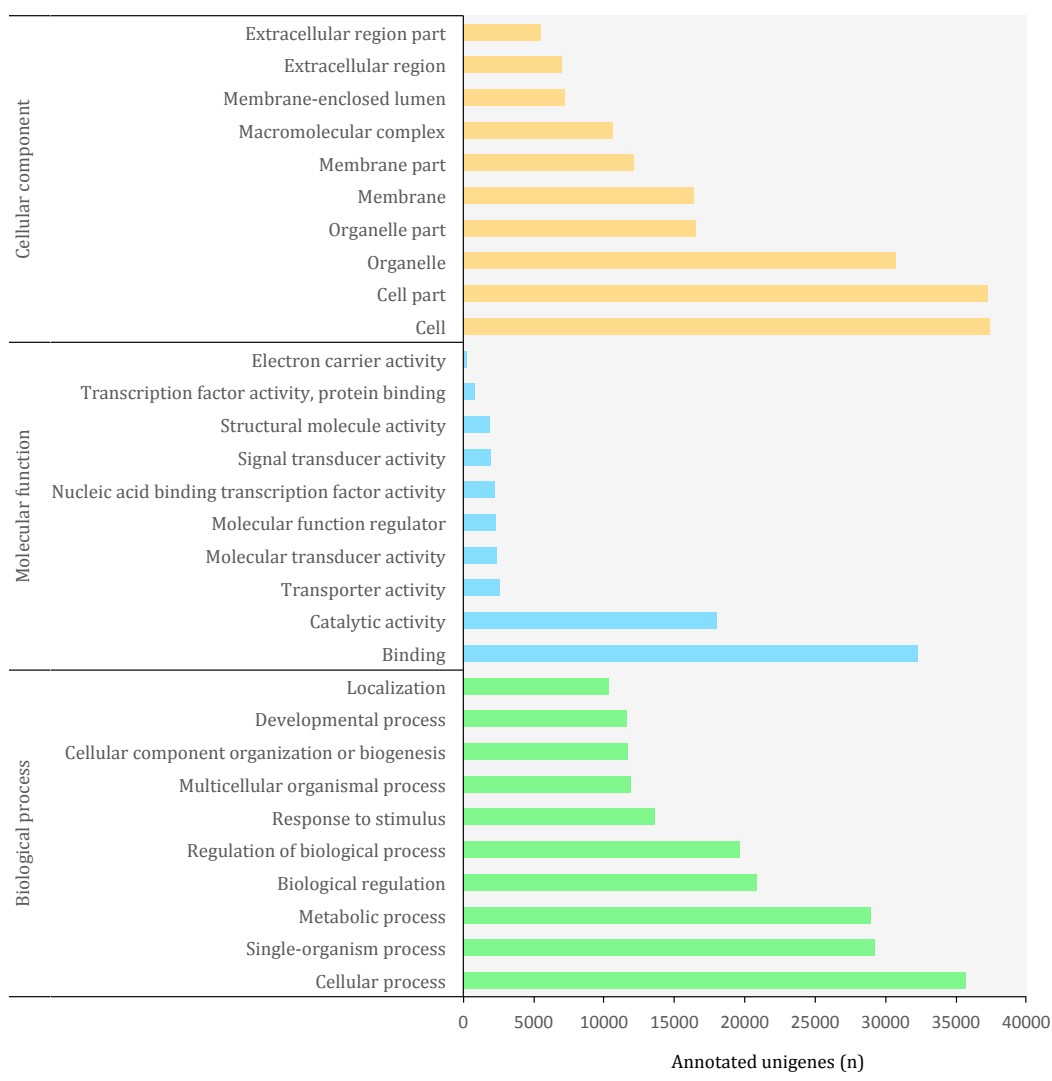

**Figure 5** **Distribution of top-10 gene ontology GO terms in the transcriptome of *O. cruralis* identified by homology with the databases via SwissProt and Pfam. Categories shown correspond to gene ontology level 2.**

the 63 antioxidant genes present in humans (*Gelain et al., 2009*) and it might be related to the high number of protein kinase domains that we recorded earlier, as well as to the habitat of *O. cruralis*. Specimens are usually encountered in tropical rainforest leaf litter, where amphibian pathogens are common (*Pounds et al., 2006*). Antioxidant genes have previously been reported from the skin of amphibians, contributing to resistance against microorganism infection or radiation injury (*Yang et al., 2009*). However, since the transcriptome of *O. cruralis* was built from tissues of intestine, liver and spleen, our results suggest that antioxidant genes in amphibians can also be expressed in different tissues besides skin. Because *O. cruralis* is mainly a lowland Amazonian rainforests frog, it would be interesting to compare these results with closely-related species living in higher altitudes (e.g. *Oreobates ayacucho*), where the temperature is lower and microbial activity too.

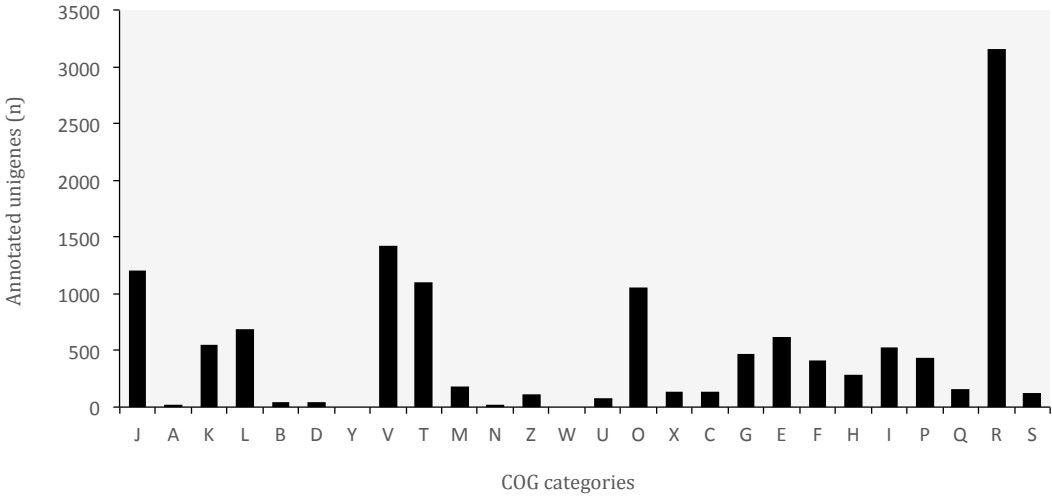

**Figure 6** **Distribution of Clusters of Orthologous Groups (COG) categories in the transcriptome of
*O. cruralis*.** J, Translation, ribosomal structure and biogenesis; A, RNA processing and modification; K,
Transcription; L, Replication; recombination and repair; B, Chromatin structure and dynamics; D, Cell
cycle control, cell division, chromosome partitioning; Y, Nuclear structure; V, Defense mechanisms; T,
Signal transduction mechanisms; M, Cell wall/membrane/envelope biogenesis; N, Cell motility; Z, Cy-
toskeleton; W, Extracellular structures; U, Intracellular trafficking, secretion, and vesicular transport; O,
Posttranslational modification, protein turnover, chaperones; X, Mobilome: prophages, transposons; C,
Energy production and conversion; G, Carbohydrate transport and metabolism; E, Amino acid transport
and metabolism; F, Nucleotide transport and metabolism; H, Coenzyme transport and metabolism; I,
Lipid transport and metabolism; P, Inorganic ion transport and metabolism; Q, Secondary metabolites
biosynthesis, transport and catabolism; R, General function prediction only; S, Function unknown.

## COG classification

The database of Clusters of Orthologous Groups (COGs) is another common tool for
functional annotation (*Galperin et al., 2015*). In this database, orthologous genes from 722
prokaryote genomes are grouped according to their biological function. The current version
consists of 4,632 COGs classified into 26 functional categories. The EggNOG database is
based on the original idea of COGs and expands it to non-supervised orthologous groups
from numerous organisms, including eukaryotes and viruses (*Huerta-Cepas et al., 2016*).
We identified a total of 37,349 (8.83%) unigenes that are present in the EggNOG database.
Of these, 12,993 belonged to the COG database, corresponding to 24 functional categories
(Fig. 6). The "general function" category (3,166; 24.37%) represented the largest group,
followed by "defense mechanisms" (1,421; 10.94%). Our results showed that genes related
to defense functions may be relatively abundant in the transcriptome of *O. cruralis*,
particularly compared to the seven anurans studied by *Huang et al. (2016)* and also to *A.
mexicanum* (*Wu et al., 2013*). In both studies, only about 2% of unigenes corresponded to
defense mechanisms. Within the unigenes involved in defense mechanisms, we identified
1,163 (81.84%) that are related to Cytochrome P450 enzymes (CYPs), while only 57 of
those genes have been found in humans (*Zanger & Schwab, 2013*). CYPs are a protein
superfamily in charge of metabolizing potentially toxic compounds, such as drugs or
products of endogenous metabolism (*Fujita et al., 2004*). This large difference in the
number of genes in humans and *O. cruralis* may indicate the presence of duplicates in our

data, but it could also be associated with some degree of myrmecophagy (feeding on ants) in this group of frogs. Because the eating habits of *Oreobates* frogs have not been studied yet, protein data from strict myrmecophagous species (e.g., poison dart frogs in the family Dendrobatidae) are needed to confirm these results.

### KEGG pathways

In the KEGG (Kyoto Encyclopedia of Genes and Genomes) database, genes from completely sequenced genomes are linked to higher-level systemic functions of the cell, the organism and the ecosystem (*Kanehisa & Goto, 2000*). Molecular-level functions are stored in the KO (KEGG Orthology) database, where each KO is defined as a functional ortholog of genes and gene products (*Kanehisa et al., 2016*). We identified a total of 38,120 (9.01%) unigenes from *O. cruralis* in the KEGG database. Of these, 25,619 unigenes have orthologs in the KO database. Many unigenes were classified under the category of organismal systems (3,704; 29.32%), followed by metabolism (3,580; 28.34%), environmental information processing (2,678; 21.20%), cellular processes (1,535; 12.15%) and genetic information processing (1,135; 8.99%) (Fig. 7). We found the largest number of unigenes to be related with signal transduction (2,241) within the category of environmental information processing. Particularly, the PI3K-Akt signaling pathway was the most frequent (184; 8.21%) among the signal transduction unigenes, followed by the MAPK signaling pathway (152; 6.78%). Both the PI3K-Akt and the MAPK signaling pathways play a major role in the development of immune cells (*Liu, Shepherd & Nelin, 2007*; *Juntilla & Koretzky, 2008*). Interestingly, the immune system category was also highly enriched (1,057 unigenes) and within the immune category, the chemokine signaling pathway comprised the highest number of unigenes (105; 9.93%). Chemokine receptors associate with G proteins to promote signaling cascades, including MAPK pathways, that cause immune responses such as degranulation, a cellular process that releases antimicrobial cytotoxic molecules to destroy invading microorganisms (*Murdoch & Finn, 2000*). This suggests that, compared to other genes, those related to the immune system are relatively abundant in the transcriptome of *O. cruralis*. We hypothesize that tropical conditions, in which high temperature and humidity are constant throughout the year, impose a crucial challenge to amphibian fitness. Although based on a single transcriptome our results lack of statistical power, this study provides a first view towards the understanding of gene evolution in Neotropical amphibians.

## CONCLUSIONS

Although large genome size renders complete genome sequencing practically unfeasible in many species, such as most amphibians, transcriptome sequencing represents a cost-effective alternative to obtain a large amount of genome-wide data. This can allow the study of selection and adaptation in natural populations, but it will also lead to advances in the study of ecological and evolutionary processes beyond the limits imposed by the use of small panels of markers. In this study, we have provided and discussed a pipeline that covers the basic elements needed to build a *de-novo* transcriptome from RNA-seq data of non-model organisms for which sequencing and assembling a genome is not a practical option. We

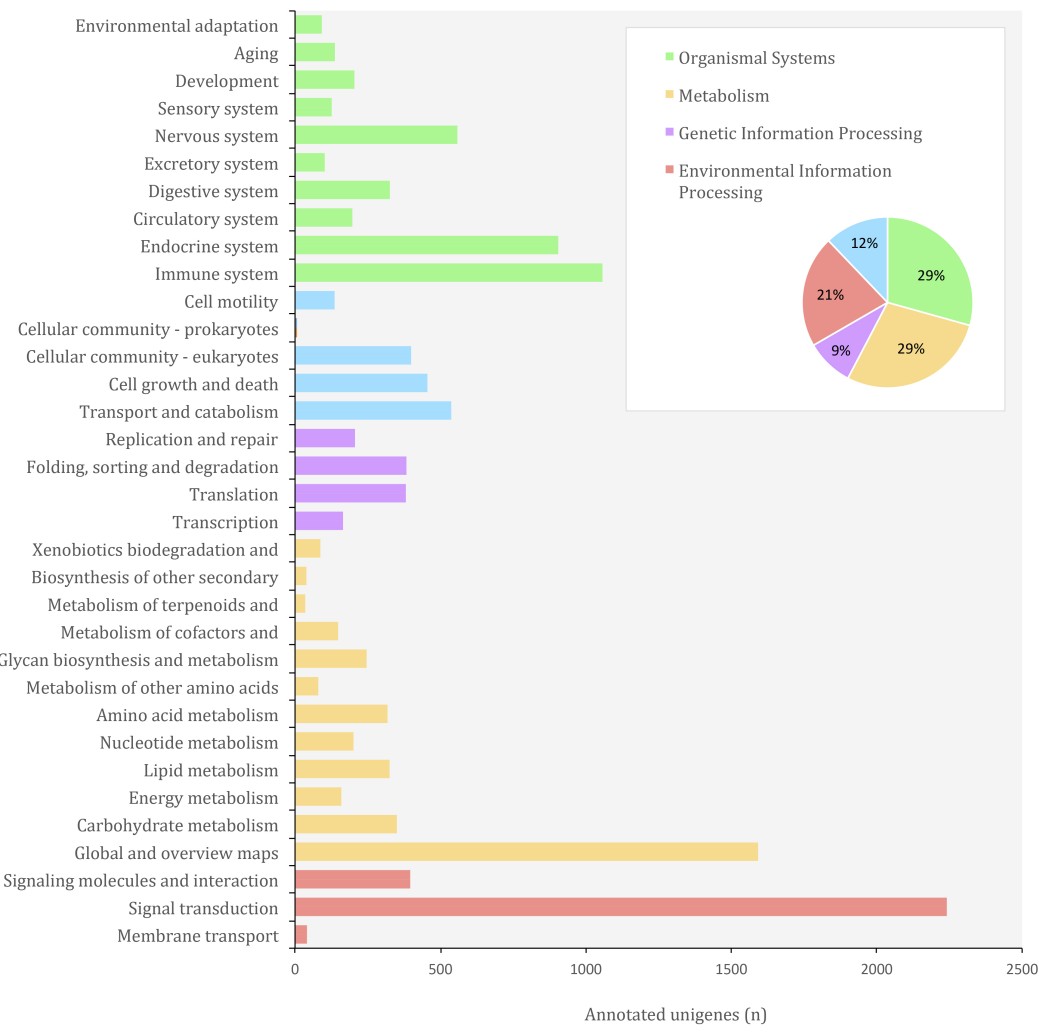

**Figure 7  Distribution of KEGG Orthology (KO) categories in the transcriptome of *O. cruralis*.**

have successfully applied this pipeline to obtain the transcriptome profile of *Oreobates cruralis*, a poorly known Neotropical frog. The data obtained here has some limitations: the specimen was unsexed, and only three tissues and one life stage were represented. Thus, the results should be taken with caution in the context of sex-specific gene expression. Nevertheless, this is the first transcriptome data available for a South American amphibian, and therefore, a stepping-stone towards the study of the diversification patterns across this group of vertebrates using genomic approaches. Once a reference transcriptome is available, capture-based approaches can help to obtain homologous sequences for a large array of closely related species at a reduced cost. In this regard, this transcriptome will serve as a valuable resource for the inference of orthologous sequences in closely related species. This, for example, will allow solving phylogenomic relationships among the species of the genus *Oreobates*, as well as studying population differentiation, demographic history and gene evolution for the different species.

## ACKNOWLEDGEMENTS

The tissue samples used for this study were provided by the frozen tissue collection of the Museo Nacional de Ciencias Naturales (MNCN-CSIC) in Madrid, Spain (MNCN/ADN collection). Sequencing was performed by the SNP&SEQ Technology Platform in Uppsala, Sweden. The facility is part of the National Genomics Infrastructure (NGI) Sweden and Science for Life Laboratory. The SNP&SEQ Platform is also supported by the Swedish Research Council and the Knut and Alice Wallenberg Foundation. Computations were performed on resources provided by SNIC through Uppsala Multidisciplinary Center for Advanced Computational Science (UPPMAX) under Project b2015409. We thank all the members of the Conservation and Evolutionary Genetics Group, as well as Dr. José Manuel Padial for constructive comments and support in the study. We also thank Anna Olsson for laboratory support.

### Funding

This work was supported by grants from the Spanish Government (Ministerio de Economía y Competitividad) to CV (CGL2013-47547-P) and to Ignacio De la Riva (CGL2011-30393), as well as a "FPI" (Formación de Personal Investigador) fellowship (BES-2014-069006) and a travel grant (EEBB-I-2016-10576) to Santiago Montero-Mendieta. The funders had no role in study design, data collection and analysis, decision to publish, or preparation of the manuscript.

### Grant Disclosures

The following grant information was disclosed by the authors:
Spanish Government (Ministerio de Economía y Competitividad): CGL2013-47547-P, CGL2011-30393.
"FPI" (Formación de Personal Investigador) fellowship: BES-2014-069006.
Travel grant: EEBB-I-2016-10576.

### Competing Interests

The authors declare there are no competing interests.

### Author Contributions

- Santiago Montero-Mendieta conceived and designed the experiments, performed the experiments, analyzed the data, wrote the paper, prepared figures and/or tables, reviewed drafts of the paper.
- Manfred Grabherr and Henrik Lantz wrote the paper, reviewed drafts of the paper.
- Ignacio De la Riva and Jennifer A. Leonard contributed reagents/materials/analysis tools, wrote the paper, reviewed drafts of the paper.
- Matthew T. Webster conceived and designed the experiments, wrote the paper, reviewed drafts of the paper.
- Carles Vilà conceived and designed the experiments, performed the experiments, wrote the paper, reviewed drafts of the paper.

## Animal Ethics

The following information was supplied relating to ethical approvals (i.e., approving body and any reference numbers):

The Spanish Government (Ministerio de Economía y Competitividad) provided full approval for the original study involving the handling of live animals.

## Field Study Permissions

The following information was supplied relating to field study approvals (i.e., approving body and any reference numbers):

The Dirección General de la Biodiversidad, Ministerio de Medio Ambiente y Agua, La Paz, Bolivia approved the field permits for the original field work.

## DNA Deposition

The following information was supplied regarding the deposition of DNA sequences:

Raw RNA-seq data in FASTQ format has been deposited at the NCBI Sequence Read Archive database (SRA) under the accession SRP106442. The transcriptome assembly in FASTA format has been deposited at DDBJ/EMBL/GenBank under the accession GFNJ00000000. The version described in this paper is the first version, GFNJ01000000. All the data is available at NCBI BioProject under the accession PRJNA384528.

## Data Availability

NCBI BioProject: PRJNA384528.

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
