# Peer review of "A practical guide to build de-novo assemblies for single tissues of non-model organisms: the example of a Neotropical frog"

_PeerJ, doi:10.7717/peerj.3702_

## Round 0.1 · original submission · Minor Revisions

I have now received two reviews of your frog transcriptome paper and both reviewers felt your paper will make a solid contribution to the literature. However, both have a number of issues/suggestions for you to deal with before your paper can be further considered. I appreciate your already having submitted the data to GenBank in both raw and assembled form. Good luck with your revisions.

Reviewer 1 ·

Basic reporting

The English looks good to me.

Experimental design

The information on the frog is insufficiently poor. Species identification remains a mystery and what I find also very strange is the missing sex information on the sampled individual frog. This is of very high relevance due to sex-specific gene expression. Another concern is that RNA has been extracted from five tissues (intestine, liver, spleen, heart and skin) but only of three (intestine, liver, spleen) transcriptomic information could be generated. Needless to say that gonadal and brain tissue would be of high relevance.

Validity of the findings

That RNA comes only from three tissues of an apparently unsexed frog is a major drawback and should be openly discussed as negative point. Therefore, I strongly recommend a change of the title "transcriptomic information from three tissues of an unsexed individual of Oreobates cruralis".

Additional comments

all covered by comments on 2 and 3; it is unclear however, if this will be sent to the author, actually.

on 2) That RNA comes only from three tissues of an apparently unsexed frog is a major drawback and should be openly discussed as negative point. Therefore, I strongly recommend a change of the title "transcriptomic information from three tissues of an unsexed individual of Oreobates cruralis".
on 3) The information on the frog is insufficiently poor. Species identification remains a mystery and what I find also very strange is the missing sex information on the sampled individual frog. This is of very high relevance due to sex-specific gene expression. Another concern is that RNA has been extracted from five tissues (intestine, liver, spleen, heart and skin) but only of three (intestine, liver, spleen) transcriptomic information could be generated. Needless to say that gonadal and brain tissue would be of high relevance.

·

Basic reporting

There are places where the text doesn't provide the detail and clarity required for the non-specialist audience that the authors have targeted. Also, the overall appearance still feels a bit like a draft and would benefit from a round of refining language and increasing the finish and quality of the figures. Figure legends could use some additional work to make them stand alone by removing phrases in inverted commas which rely on the main text for clarity.

Experimental design

The genomic data will provide an excellent resource for future studies using a number of different approaches and the authors appear to have been thorough and careful in their analyses, applying suitable tools appropriately.

Validity of the findings

no comment

Additional comments

Montero-Mendieta et al present the transcriptome of a South American frog which they use to outline a workflow for processing RNAseq data from non-model organisms. The genomic data will provide an excellent resource for future studies using a number of different approaches and the authors appear to have been thorough and careful in their analyses, applying suitable tools appropriately. However I felt the manuscript itself needed some more work before it would be ready for publication. In the main there are places where the text doesn't provide the detail and clarity required for the non-specialist audience that the authors have targeted. Also, the overall appearance still feels a bit like a draft and would benefit from a round of refining language and increasing the finish and quality of the figures.

Figure legends could use some additional work to make them stand alone by removing phrases in inverted commas which rely on the main text for clarity.

abstract the abstract suggests that this will be more of a methods paper which will serve a community of non-bioinformaticians wanting to conduct genomics studies with non-model organisms, and that your frog study is proof of efficacy of the pipeline. The abstract reflects the article well. In contrast, the title suggests that the frog is front and centre. introduction/justification for using RNAseq here in the abstract could probably be more concise?

141 define m.a.s.l
157 define RIN
189-193 please state why you did two rounds of normalisation prior to assembly
209 settings?
220-234 Given that a major goal of this manuscript is to guide non-specialists who are working with non-models I think this whole section lacks detail of what was being done and why
257 scientific numbers would be easier to digest
306-8 I've don't know how this result compares to other studies. It definitely looks like BUSCO is a really useful resource for this type of analysis but my impression is that CEGMA has been the default tool previously for making such assessments of assembly completeness. Since it's possible to compare to other studies in this way it might be worth presenting the BUSCO results alongside CEGMA?
328 consists OF
329 stick to past tense
331 "WITHIN a reference database" or "blast searches AGAINST a reference database to find..."?
331 the collection of...
337-341 confusing!
364 I think you could bolster the interest of this "domain XX had the most hits, domain YY had the second most" approach. Are you able to show that any domains are enriched relative to any other animal group and therefore might have interesting outcomes for the biology of this species or class?
Fig 5 text in this figure is probably too small
Fig 6 I think this data would be more effectively displayed in a table where the numbers could be more intimately associated with the categories that they represent.

---

## Round 0.2 · Minor Revisions

You have done a solid job taking care of most of the critiques from the previous round of reviews. I have sent your article to the more critical of the first two reviewers and s/he is still concerned about the title and the associated lack of determination of sex for the organism studied. I agree with these concerns. This reviewer has proposed an alternative title, which I like and think is appropriate. Frankly, I think it gives your paper broader appeal.

Please also add a critical discussion of the limitation of your study due to the lack of a definitive sex of your frog in the context of sex-specific gene expression. I think all this works reasonably well into the context of the non-model organism (things are not ideal).

Once these adjustments are made, please upload your revision and a cover letter and then I'll take a look, but not send it out for further review.

Reviewer 1 ·

Basic reporting

Also the new title is really more than overstated. How can a specimen that was unsexed and of which only three tissues have been analysed become "First transcriptome assembly". I can only hope that the editor will not permit this paper to be published with this title. I propose another time one:
A practical guide to build de-novo assemblies for single tissues of non-model organisms: The example of an unsexed Neotropical frog.

Experimental design

lines 153/154: "The small size of the specimen suggests that it was likely to be a female." cannot stay like this in a scientific paper. Juvenile males are also of small size... :(
Correct this:
"The specimen was unsexed."

Validity of the findings

The missing sex information and the resulting limitations from this due to sex-specific gene expression should be critically discussed

Additional comments

The title is still not really appropriate from my perspective...

---

## Round 0.3 · accepted · Accept

Thank you again for your careful revision and adding acknowledgements to the limitations of your study in the text. I'm happy with your modification of the title. I think your paper is now ready to go.